# Construction of Engineered Muscle Tissue Consisting of Myotube Bundles in a Collagen Gel Matrix

**DOI:** 10.3390/gels9020141

**Published:** 2023-02-08

**Authors:** Kazuya Furusawa, Yuuki Kawahana, Ryoya Miyashita

**Affiliations:** Department of Applied Chemistry and Food Science, Faculty of Environmental and Information Science, Fukui University of Technology, Fukui 910-8505, Japan

**Keywords:** tissue engineering, engineered muscle tissue, collagen hydrogels, viscoelasticity

## Abstract

Tissue engineering methods that aim to mimic the hierarchical structure of skeletal muscle tissue have been widely developed due to utilities in various fields of biology, including regenerative medicine, food technology, and soft robotics. Most methods have aimed to reproduce the microscopical morphology of skeletal muscles, such as the orientation of myotubes and the sarcomere structure, and there is still a need to develop a method to reproduce the macroscopical morphology. Therefore, in this study, we aim to establish a method to reproduce the macroscopic morphology of skeletal muscle by constructing an engineered muscle tissue (EMT) by culturing embryonic chicken myoblast-like cells that are unidirectionally aligned in collagen hydrogels with micro-channels (i.e., MCCG). Whole mount fluorescent imaging of the EMT showed that the myotubes were unidirectionally aligned and that they were bundled in the collagen gel matrix. The myotubes contracted in response to periodic electrostimulations with a frequency range of 0.5–2.0 Hz, but not at 5.0 Hz. Compression tests of the EMT showed that the EMT had anisotropic elasticity. In addition, by measuring the relaxation moduli of the EMTs, an anisotropy of relaxation strengths was observed. The observed anisotropies could be attributed to differences in maturation and connectivity of myotubes in the directions perpendicular and parallel to the long axis of the micro-channels of the MCCG.

## 1. Introduction

Tissue engineering of skeletal muscles has been extensively investigated to provide model systems for studying the differentiation mechanisms of muscle tissues and to be used for regenerating severely damaged muscles [1,2,3]. In addition, engineered skeletal muscle tissue produced using animal cells has been regarded to be a potential alternative to meat, referred to as “cultured meat” [4,5,6]. In the field of robotics, engineered skeletal muscles have been used as living actuators for use in the construction of organic soft robotic structures [7,8,9].

In native muscle tissues, such as skeletal muscles and the heart, muscle cells align parallel to the long axis of the muscle, and this unidirectional alignment results in the contraction function; therefore, there have been various methods developed for reproducing the unidirectional orientation of myogenic cells, such as myoblasts and myotubes [9,10,11]. However, native muscles have more complex hierarchical structures than the simple orientation of myogenic cells, and there have been no methods developed for reproducing these structures.

In myocytes (muscle fibers), actin and myosin form an ordered structure, which is known as the sarcomere structure. This hierarchical structure has important roles in muscle actuation and the texture of meat. Therefore, a method for reproducing a bundle of muscle fibers is a potential strategy for providing organic actuators with efficient actuating functions and cultured meat with textures of native meats.

Previously, we prepared collagen hydrogels with multi-channel structures (namely, multi-channel collagen gel (MCCG)) [12,13]. The multi-channel structures mimicked endomysium structures, and therefore, they could be used to reproduce the hierarchical structure of fasciculus [14]. In this study, the aims were to develop a method for reproducing the hierarchical structure and the mechanical properties of skeletal muscle by using MCCG. For these purposes, we constructed engineered muscle tissue (EMT) by culturing chicken embryonic muscle cells (CEMCs) in genipin-crosslinked MCCG. Changes in the morphology of the EMT were investigated using whole mount fluorescent imaging. In addition, the anisotropic mechanical properties of the EMT were investigated by measuring the stress–strain curve and stress relaxation, and then the results were compared with those of native chicken skeletal muscle.

## 2. Results and Discussion

### 2.1. Construction Method of the EMT Using MCCG

To construct the EMT samples, we obtained cells from the leg muscles of a chicken embryo on Day 11 (namely, chicken embryonic muscle cells (CEMCs)). Some of the CEMCs cultured in a myogenic differentiation medium showed cell fusion and formed a myotube-like morphology. The myotube-like cells also showed autonomous contractions (as shown in Appendix A). The results showed that the CEMCs used in this study had myogenic differentiation properties. Because the fraction of cells with myogenic differentiation properties decreased with increased passages, in this study, we used fewer than five passages during the harvest.

Engineered muscle tissues were constructed by seeding CEMCs on MCCG. For efficient cell seeding, the culture vessel containing the cells and the scaffold were rotated at 85 rpm using a rotary shaker. The seeded cells mostly adhered to the surface of the MCCG and clogged the micro-channels; no cells adhered to the surface of the polystyrene dish. Daily observations of the samples showed that CEMCs in the micro-channels proliferated in the growth medium. During culturing, some of the cells moved from the EMT samples to the surface of the polystyrene dish, and then proliferated. On Day 7 (for a sample thickness of 1.5 mm) or Day 9 (for a sample thickness of 5 mm), the surfaces of the MCCGs and the bottom of the polystyrene dish were fully covered with CEMCs. Therefore, on Day 7 and Day 9, we exchanged the growth medium for the differentiation medium to induce the differentiation of CEMCs.

### 2.2. Hierarchical Structures of the EMT

Figure 1 shows the whole mount fluorescence microscopy images for an EMT sample at Days 3, 14, and 32. On Day 3, the cells proliferated in the micro-channels and formed cell aggregates with the shape of the micro-channels, which is shown in Figure 1A. The upper panels in Figure 2 show that the cell aggregates consisted of isotropically entangled cells. In general, confined spaces, such as micropatterned surfaces and micro-channels, affect cell morphology [15,16]. Similarly, the micro-channels of the MCCGs could have confinement effects on the cell morphology. However, it was not observed at the scale of cell size. Because the diameters of the micro-channels were larger than those of the CEMCs, confinement effects were not observed in the morphology of the cells. Several micro-channels were completely clogged with cell aggregates, whereas there were also void micro-channels. We cut both sides of the MCCG to open the micro-channels, but several micro-channels remained closed. Therefore, we could not completely fill all micro-channels with cells by initial seeding, which was a limitation of our method. By contrast, void micro-channels could play the role of transportation ducts for nutrients, oxygens, and waste products. The upper panels of Figure 2 show that several cells in a micro-channel migrated to the next micro-channel. Because the cells that migrated into the void micro-channel could proliferate and form cell aggregates, all micro-channels might be filled with the cells by further culturing. In fact, Figure 1 shows that the number of void micro-channels decreased with culture time, and most micro-channels were filled with cells by Day 32. The myotubes in the MCCG were long and narrow. In addition, no cross striations were observed. Therefore, the myotubes in the MCCG were not fully differentiated into myofibers. By contrast, the myotubes in the MCCG were bundled in the collagen gel matrix that consisted of collagen fibers. Because the perimysium and endomysium also consisted of collagen fibers, the EMT sample partially mimicked the macroscopic morphologies of skeletal muscle.

On Day 14, the cells in the micro-channels formed into large cells that were elongated and parallel to the long axis of the micro-channel, which is shown in Figure 1B. In addition, the mid-panels in Figure 2 show that the large cells were multinucleated cells. This result shows that the CEMCs differentiated into myotubes by cell fusion. The myotube-like cells were centered in a micro-channel and had lower expression levels of F-actin than those of the myotube-like cells near the surface of the micro-channel. The cells with the lower expression levels of F-actin could be non-differentiated CEMCs. In the center of the cell aggregates observed on Day 3, the concentrations of nutrients, such as glucose and growth factors, were lower than those at the periphery. In general, myoblast differentiation is induced by nutrient starvation [17,18,19,20]. Therefore, the differences in differentiation of CEMCs in the micro-channels might be attributed to the concentration gradients of nutrients. At Day 32, the number of myotube-like cells increased, and they mostly filled the micro-channels of the MCCG, which is shown in Figure 1C and the lowest panels of Figure 2.

### 2.3. Contractile Functions of Myotubes in the EMT

By daily observation of the cells in the MCCG, we found that the cells in the micro-channels exhibited autonomous contractions, which are shown in Appendix A. This clearly showed the myotube differentiation of the CEMCs. Appendix A shows that the myotubes in the EMT sample on Day 14 contracted in response to periodic electrostimulation with frequencies from 0.5 to 2.0 Hz, but not at 5.0 Hz. Figure 3 shows the time course displacement of engineered muscle tissue during the electrostimulation.

The displacement periodically changed with time, and it synchronized with the periodic electrostimulation. Figure 4 shows the results of the Fourier transform applied to data recorded in the time ranges 27~67 s, 67~108 s, and 108~149 s, in Figure 3, respectively. Peaks were observed at 0.5, 1.0, and 2.0 Hz, as shown in Figure 4. The peak frequencies were in agreement with the frequencies of applied electrostimulation and, therefore, indicated synchronous contractions of the myotubes in the EMT sample. By contrast, no peaks were observed at 5.0 Hz in any Fourier analyses. The myotubes in the motion tracing area may not have been able to respond to the electrostimulation at 5.0 Hz. The electrically induced contractions of myotubes were not completely simultaneous, and there were also myotubes with delayed contractions. The gel matrix of MCCG did not undergo detectable deformation by the electrically induced contractions of myotubes. As shown in Figure 2, the myotubes were discretely distributed in the micro-channels of MCCG, and they did not connect from end to end of the EMT. This result suggests that the contraction force evoked by myotubes was localized and was not transmitted to the whole EMT. Therefore, macroscopic deformation of the EMT was not observed.

### 2.4. Stress–Strain Curves of EMT

To investigate the effects of the macroscopically anisotropic structures on the mechanical properties of the EMT samples, we measured the anisotropy of the stress–strain curves and the stress relaxation behaviors. The samples were compressed in directions perpendicular and parallel to the long axis of the micro-channels of the EMT and MCCG or the orientation of the myofibers of the CT. Hereafter, the strain directions are abbreviated by “perpendicular direction” and “parallel direction”, respectively. Figure 5A,B show the stress–strain curves for the EMT and the MCCG deformed in the perpendicular and parallel directions, respectively. The stress linearly increased with strain in the measured range, irrespective of the deformation directions. These results show that the stress–strain curves obey Hooke’s law. Figure 5C shows the elastic moduli for the EMT and MCCG in the perpendicular and parallel directions. The elastic modulus of EMTs in the perpendicular direction was significantly higher than that in the parallel direction. This result shows that the EMT has anisotropic elasticity. In addition, the elastic modulus of EMTs was significantly higher than that of MCCG, irrespective of the deformation direction. In contrast, there was no anisotropic elastic modulus of MCCG between the perpendicular and the parallel directions. The results show that culturing CEMCs has an effect on the anisotropic mechanical properties of the EMT. Next, we measured the anisotropic elastic modulus of the CT. The stress–strain curves and the anisotropic elastic modulus are shown in Appendix A. Although the results were highly variable, and there was no significant difference in the parallel direction, the elastic modulus in the parallel direction was higher than that in the perpendicular direction, three times out of four. Therefore, the mechanical properties of the EMT did not reproduce those of native tissue. Because, in the parallel direction, the myotubes and undifferentiated CEMCs could not be connected completely, cell sliding occurred in the parallel direction due to the compression. By contrast, cell sliding did not occur due to the compression exerted in the perpendicular direction. Therefore, the elastic modulus of the EMT in the parallel direction was less than that in the parallel direction.

### 2.5. Stress Relaxation Behaviors of EMT

To further investigate the effects of the macroscopically anisotropic structure on the viscoelastic properties of the EMT, we measured the relaxation modulus. Figure 6 shows the relaxation moduli of the EMT in the perpendicular and parallel directions. The relaxation modulus consisted of fast and slow relaxations, and converged to the equilibrium modulus, irrespective of deformation directions. Therefore, to analyze the viscoelastic properties of the samples, the relaxation moduli were fitted by using the following equation assuming the effect of the distribution of relaxation times with slight modification [21]:
(1)$$E\left(t\right)=E_{1}e^{-{\left(\frac{t}{\tau_{1}}\right)}^{\beta }}+E_{2}e^{-{\left(\frac{t}{\tau_{2}}\right)}^{\gamma }}+E_{\mathrm{eq}}\;$$
where $E_{1}$ and $E_{2}$ are, respectively, the relaxation strengths of the fast and slow relaxations; $E_{\mathrm{eq}}$ is the equilibrium elastic modulus; $\tau_{1}$ and $\tau_{2}$ are the relaxation times of the fast and slow relaxations; and β and γ are indexes related to the polydispersity of relaxation times for the fast and slow relaxations. If β and γ are 1, the stress relaxation is expressed by a simple relaxation. By contrast, low values of β and γ (0<β,γ<1) indicate that the distribution of relaxation times is wide. The instantaneous elastic modulus ($E_{0}$) is given by the following equation:(2)$$E_{0}=E_{1}+E_{2}+E_{\mathrm{eq}}$$

The relaxation moduli were effectively expressed using Equation (1) which is shown in Appendix A. By curve fitting, we obtained the relaxation parameters associated with the deformation applied to both the perpendicular and the parallel directions, as shown in Figure 7. In this study, the stress–strain curve was determined by measuring the stress at 30 s after stepwise deformation. Therefore, the elastic modulus calculated from the stress–strain curve should be comparable to $E_{\mathrm{eq}}$. In fact, the elastic moduli of the EMT shown in Figure 5C were comparable to the values of the $E_{\mathrm{eq}}$ in Figure 7. However, there were no significant differences between the values of the equilibrium elastic moduli ($E_{\mathrm{eq}}$) measured during the two different deformation experiments. The result appears to be inconsistent with those related to the elastic moduli calculated from the stress–strain curve. However, this inconsistency was attributed to the presence of slow relaxation at 30 s after stepwise deformation, because the relaxation time of slow relaxation was approximately equal to 20 s. Therefore, the contribution of slow relaxation moduli is also included in the results shown in Figure 5. The relaxation strengths of the fast and slow relaxations in the perpendicular direction were significantly higher than those in the parallel direction.

By contrast, there were no significant differences in the relaxation times and the polydispersity indexes of the relaxation phenomena that occurred along the perpendicular and parallel directions, respectively. The relaxation time (τ) is related to the elastic modulus (E) and the viscosity (η) of materials by the following equation:(3)$$\tau =\frac{\eta }{E}$$

Equation (3) shows that an increase in the elastic modulus results in a decrease in the relaxation time, whereas an increase in the viscosity prolongs the relaxation time. If either the elasticity or the viscosity is different between both directions, a different relaxation time should be observed. However, the relaxation times ($\tau_{1}$ and $\tau_{2}$) in both directions were comparable, whereas the relaxation strengths ($E_{1}$ and $E_{2}$) in the perpendicular direction were significantly higher than those in the parallel direction. Therefore, the results suggest that elasticity and viscosity in the perpendicular direction are both proportionally larger than those in the parallel direction. As speculated above, the deformation in the parallel direction induces the sliding cells, which can be accompanied by simultaneous decreases in elasticity and viscosity to maintain the ratio between relaxation times at nearly constant. We suggest that the differences are attributed to the anisotropic microscopic deformations in the MCCG. We also measured the relaxation moduli of the MCCG and CT. The relaxation moduli of the MCCG and CT are shown in Appendix A, respectively, and the results of fitting the relaxation moduli with Equation (1) are shown in Appendix A, respectively. The relaxation parameters are comparatively shown in Appendix A. There are no significant differences in $\tau_{1}$, $\tau_{2}$, β, and γ concerning the two directions of deformation. Instead, $E_{1}$, $E_{2}$, $E_{eq}$, and $E_{0}$ of the EMT in the perpendicular direction are significantly higher than those for the MCCG, whereas there are no significant differences in $E_{1}$, $E_{2}$, $E_{eq}$, and $E_{0}$ between the EMT and the MCCG in the parallel direction. The results show that culturing CEMCs mainly enhances the elastic moduli and viscosity coefficients in the perpendicular direction. $E_{1}$, $E_{2}$, and $E_{0}$ of the EMT in the perpendicular direction are significantly higher than those of the CT. Although they are not significantly different, the parameters related to the relaxation strength of the EMT in the parallel direction are lower than those of the CT. The results indicate that the myogenesis of the EMT in the parallel direction is immature. Therefore, to reproduce the viscoelastic properties of skeletal muscle, in a future study, we have to develop an adequate method for maturing the EMT in the parallel direction.

## 3. Conclusions

In this study, we developed a method for constructing EMTs where the myotubes were unidirectionally aligned and bundled in a collagen gel matrix. The morphology of the EMT partially reproduced the macroscopic structure of skeletal muscle tissues; however, the sarcomere structure was not observed. To reproduce the hierarchical structure of skeletal muscle, a method to further promote myogenic differentiation should be developed. Because myoblast differentiation is dependent on the mechanical properties of scaffold materials and compositions of growth factors and extracellular matrices, we intend to optimize them for efficient differentiation. On the other hand, we found that myotubes in the EMT contracted in phase with the applied periodical electrostimulation with frequencies ranging from 0.5 to 2.0 Hz. The mechanical tests showed that the EMT had anisotropic viscoelastic properties. However, to reproduce the anisotropic viscoelastic properties of skeletal muscle tissue, we are concerned with improving the connectivity of cells along the longitudinal axis of myotube bundles in a future investigation.

## 4. Materials and Methods

The chicken muscle was obtained from a chicken embryo on Day 11. The muscle was minced using scissors and then incubated in 5 mg/mL type I collagenase (035-17604, Wako Co., Ltd., Tokyo, Japan) for 30 min at 37 °C. The dispersed cells were plated onto a Matrigel (356234, BD Biosciences, San Jose, CA, USA)-coated polystyrene dish with a diameter of 60 mm and cultured in DMEM/F12 (11330057, Gibco, Waltham, MA, USA) supplemented with 10% fetal bovine serum (Sigma-Aldrich, Burlington, MA, USA), 1% non-essential amino acids (11140050, Gibco), 1% penicillin–streptomycin (26253-84, Nacalai Tesque Inc., Kyoto, Japan), 10 μg/mL insulin (I9278-5ML, Sigma-Aldrich), and 20 ng/mL fibroblast growth factor (100-18B, Peprotech, Cranbury, NJ, USA), up to 90% confluence in a 5% CO_2_ incubator at 37 °C. The chicken embryonic muscle cells (CEMCs) were preserved in a deep freezer using the conventional freeze preservation method with the freeze preservation solution Bambanker^®^ (CS-02-001, GCLTEC).

To check the myogenic differentiation properties, CEMCs were plated onto a 24-well cell culture plate and cultured in a differentiation medium that consisted of DMEM with 2% horse serum, 1% NEAA, 1% penicillin–streptomycin, and 10 µg/mL insulin, for 2 weeks.

In this study, we used atelocollagen solution (IPC-50, Koken Co., Ltd., Tokyo, Japan) to prepare the MCCG. The concentration of collagen and the pH were 5 mg/mL and pH 3.0, respectively. Types of collagens in the solution were 95% type I collagen and 5% type III collagen. The MCCGs were prepared using the following procedures: First, the silicone rubber molds, shown in Appendix A, were assembled on a 60 mm polystyrene dish. Then, the collagen solution was fulfilled in the molds. Finally, a phosphate-buffered solution to prepare MCCG (PBS(g)) consisting of 20 mM Na_2_HPO_4_ and 13 mM KH_2_PO_4_ was poured into the polystyrene dish. After 12 hours, PBS(g) was removed, and then 4 mL of 5 mM genipin solution was added to chemically crosslink the MCCG. The molds were removed using sterile tweezers, and then the MCCGs were incubated at 37 °C overnight. After genipin crosslinking, the MCCG was rinsed 4 times with PBS(-) and then incubated for 5 min 4 times in PBS(-) to remove the unreacted genipin. Because the multi-channel structure of MCCG was closed, both sides were cut at 3 mm from the edge of gel using a scalpel to open the multi-channel structure. Before seeding the chick myoblast, the MCCGs were incubated in 2 mL of DMEM with 1/100 diluted Matrigel to coat the surface of the MCCGs to promote the myogenic differentiation of CEMCs [22,23].

The preserved chick myoblast was thawed and then plated onto a 60 mm polystyrene dish at 6.0 × 10^5^ cells/dish and incubated in the growth medium, which was DMEM supplemented with 10% FBS, 1% NEAA, 1% PS, 10 μg/mL insulin, and 20 ng/mL FGF up to 90% confluency. The cells were passaged onto a 100 mm polystyrene dish, 1.0 × 10^6^ cells/dish, and incubated in the growth medium up to 90% confluency. To construct the EMT, the cells were seeded into the MCCGs at 1.0 × 10^7^ cells/dish for the MCCG with 1.5 mm in thickness and at 3.0 × 10^7^ cells/dish for the MCCG with 5.0 mm in thickness. During the whole culture period, for efficient seeding of cells into the MCCGs and supplying of nutrients and oxygen to the EMT, the samples were rotated at 85 rpm by using a CO_2_-resistant rotary shaker (88881103, Thermo Fisher Scientific K. K., Waltham, MA, USA) which was placed in a 5% CO_2_ incubator (MCO-170AIC, PHC Corporation, Wood Dale, IL, USA). The samples with thicknesses of 1.5 mm and 5.0 mm were, respectively, incubated in the growth medium up to Day 7 and Day 10, and then cultured, respectively, in the differentiation medium up to Day 32 and Day 33.

To apply the electrical stimulation to the EMT, we prepared a hand-made electrode, which is shown in Appendix A. The electrode consisted of two graphite rods with diameters of 1.3 mm, which were fixed to the cap of a 60 mm polystyrene dish, and it was connected to an electrical stimulator (SEN-2201, Nihon Koden Co., Ltd., Irvine, CA, USA). The EMT with a thickness of 1.5 mm at Day 14 was stimulated by periodical electric pulses with a voltage of 100 V, duration of 3 ms, and frequencies between 0.5 and 5.0 Hz. The contraction induced by electric stimuli was recorded using a high-speed digital camera (VEX-120, Wraymer Inc., Osaka, Japan) mounted on the phase contrast microscope (CKX-53, Olympus Co., Ltd., Tokyo, Japan). The frame rate was 25 frames per second. To analyze the motion of the myotubes in the engineered muscle, we modified the obtained movie by using Image J and the following procedures: First, the color type of the movie was changed from RGB to grayscale. Then, the movie was binarized by using the “threshold” program of Image J [24]. Next, we cropped the area containing clearly contracting myotubes as a motion tracing area. Finally, by using the “analyze particle” program of Image J, the coordinates of the center of mass of the EMT (x and y) for each frames were determined. The motion of the EMT was traced by the in plane displacement (d) which was calculated by using the following equation:(4)$$d=\sqrt{{\left(x-x_{0}\right)}^{2}+{\left(y-y_{0}\right)}^{2}}\;$$
where $x_{0}$ and $y_{0}$ are the coordinates of center of mass of EMT at the initial frame (t = 0 s). The time course of d was further processed by fast Fourier transform on Excel 2013 (Microsoft Corporation).

The stress–strain curves and the stress relaxation behavior for the EMTs at Day 33, the chicken thigh (CT) purchased from a supermarket, and the MCCGs were measured by using a homemade rheometer [25]. The dimensions of the samples (thickness, width, and length) were measured by using a vernier caliper. The samples were compressed by an aluminum cylinder which was connected to a load cell (LTS-50GA, Kyowa Electronic Instruments Co. Ltd.) mounted on the automated stage actuated by a micro-actuator (AMH-15 CHUO Precision industrial Co., Ltd., Tokyo, Japan). As mentioned above, EMT has a macroscopically anisotropic structure. To investigate the anisotropic mechanical properties, the samples were compressed in directions perpendicular and parallel to the long axis of the micro-channel of the EMT and MCCG or the long axis of the myofibers of the CT. Hereafter, the compression directions are abbreviated by “perpendicular direction” and “parallel direction”, respectively. The force sensed by the load cell was recorded by a sensor interface (PCD-300B, Kyowa Electronic Instruments Co., Ltd., Tokyo, Japan). The stress was calculated by dividing the force by the cross-sectional area of the sample. The displacement of the stage was monitored by a laser reflection position sensor (LB-02, Keyence Corporation, Itasca, IL, USA). An indentation length was calculated from a force curve where the force was plotted against the displacement. The strain was determined by dividing the indentation length by the sample thickness. The stress–strain curves were determined by measuring the stress at 30 s after stepwise deformation. The elastic moduli of the samples were calculated by using Hooke’s law:(5)σ=Eγ
where σ, E, and γ are the stress, elastic modulus, and strain, respectively. Because of the measurement method of the stress–strain curve, there are no/fewer effects of stress relaxation on the elastic modulus. Therefore, the elastic modulus should be comparable to the equilibrium elastic modulus ($E_{\mathrm{eq}}$). The stress relaxation was determined by monitoring the time course of stress after deforming the sample with a constant strain. The strain for measuring the stress relaxation was equal to approximately 5%. By dividing the stress by the strain, the stress relaxation curves were expressed as the relaxation modulus E(t). All mechanical tests were performed at room temperature.

EMTs at Days 3, 14, and 32 were fixed by incubating in 2% paraformaldehyde solution at 4 °C overnight. The samples were washed 4 times with PBS(-). For whole mount imaging, the samples were incubated in a blocking solution consisting of 5% bovine serum albumin and 0.3% triton-X100 at 37 °C for 1 h. The samples were fluorescently stained with 5 U/mL Alexa Fluor^TM^ 488 phalloidin (A12379, Invitrogen) and DAPI (R37606, Invitrogen). The fluorescent image of collagen was observed by using autofluorescence, which is attributed to the genipin crosslinking [26]. The whole mount images were obtained using an all-in-one microscope (BZ-X 800, Keyence Corporation).

## Figures and Tables

**Figure 1 gels-09-00141-f001:**
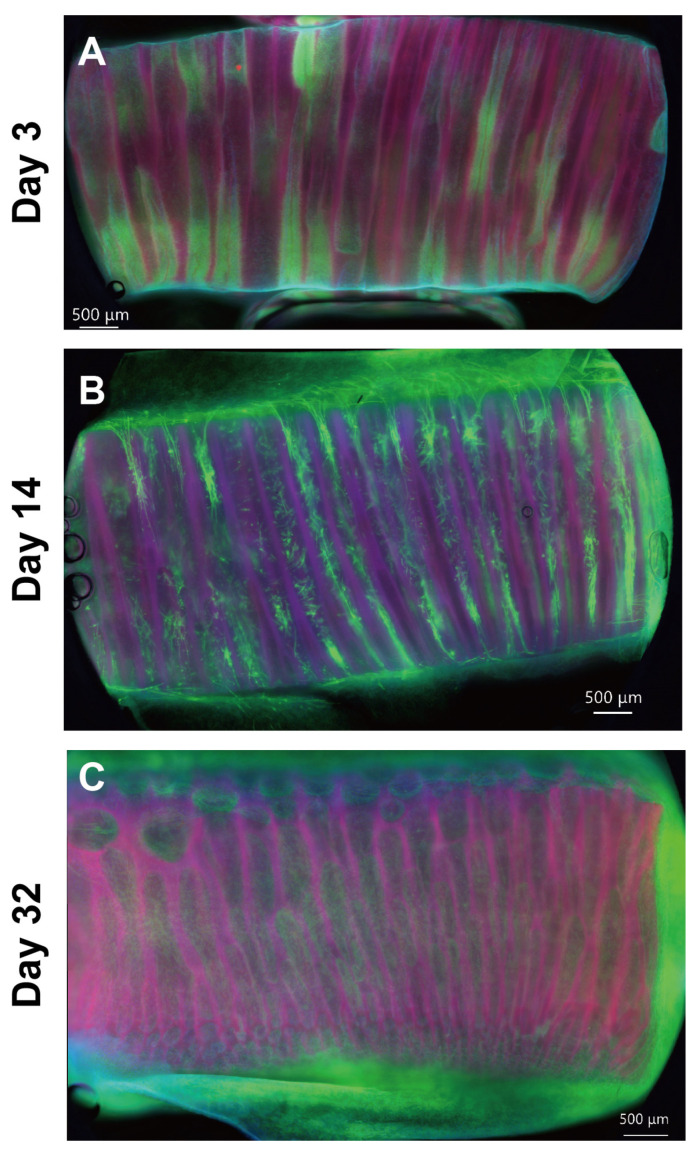
Whole mount fluorescence microscopy images of the EMT obtained at different culture periods ((**A**) 3 days, (**B**) 14 days, and (**C**) 32 days). Nuclei, F-actin, and collagen are colored blue, green, and red, respectively.

**Figure 2 gels-09-00141-f002:**
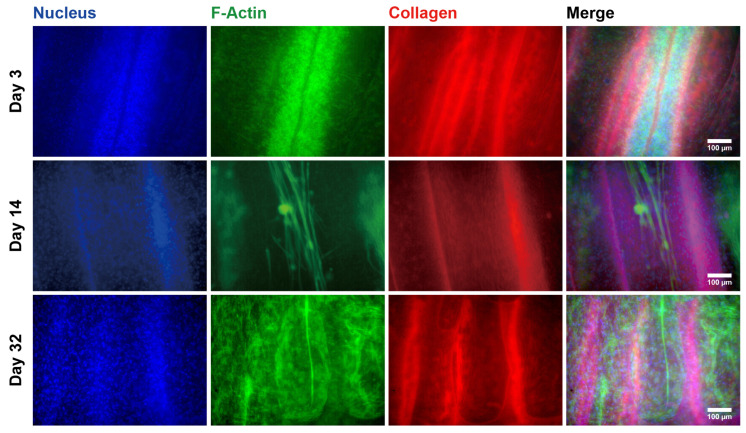
Fluorescent images of myotubes in the micro-channels of the MCCG.

**Figure 3 gels-09-00141-f003:**
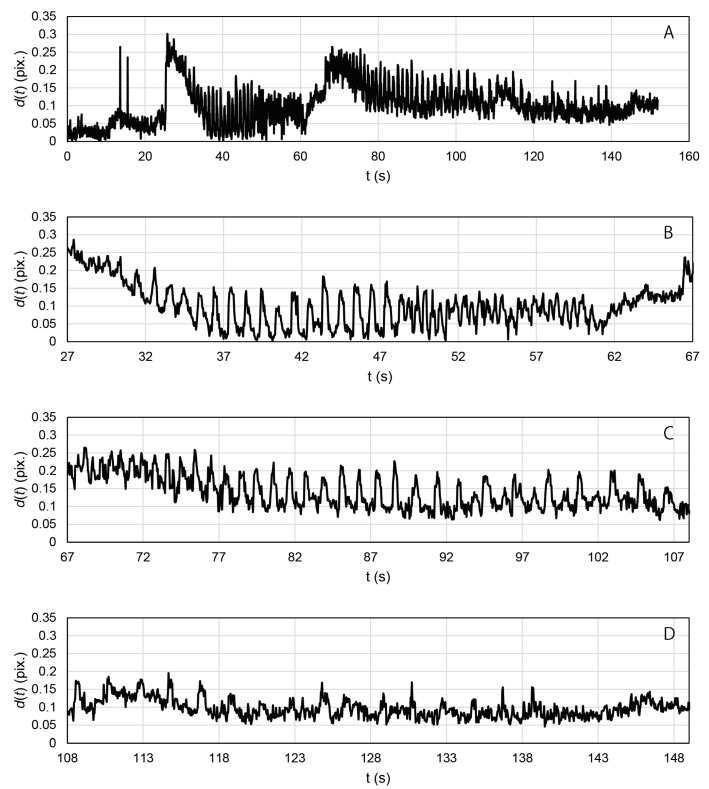
Time course displacement of the EMT (*d*(*t*)) during the periodical electrostimulation: (**A**) whole process; (**B**) 27–67 s; (**C**) 67–108 s; (**D**) 108–149 s.

**Figure 4 gels-09-00141-f004:**
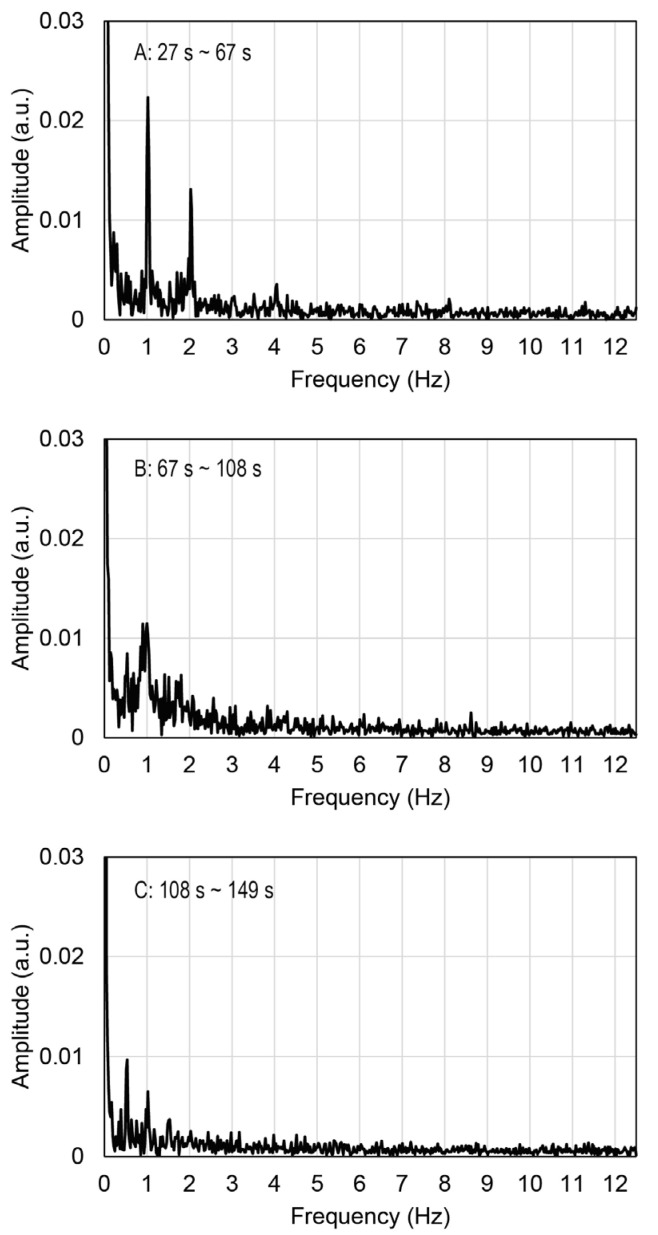
Fast Fourier transform applied to the signals illustrated in Figure 3B–D.

**Figure 5 gels-09-00141-f005:**
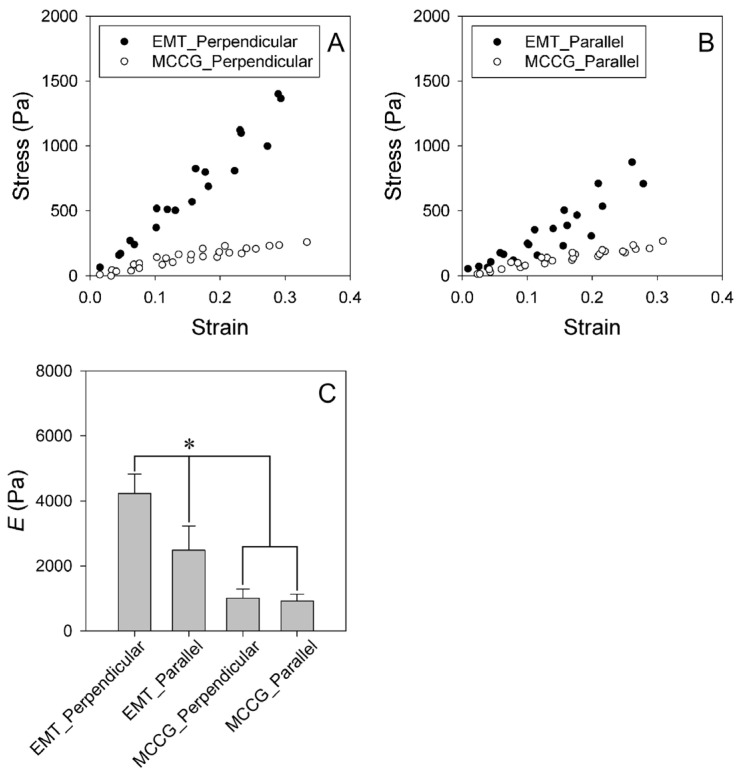
(**A**,**B**) Stress–strain curves of the EMT and MCCG in the perpendicular and parallel directions. All measured data (*n* = 4) are shown; (**C**) comparison of elastic moduli of the EMT and MCCG. Error bars indicate standard deviation (*n* = 4). The asterisk indicates significant differences of *p* < 0.05. Comparison performed by using ANOVA test and post hoc Tukey’s test.

**Figure 6 gels-09-00141-f006:**
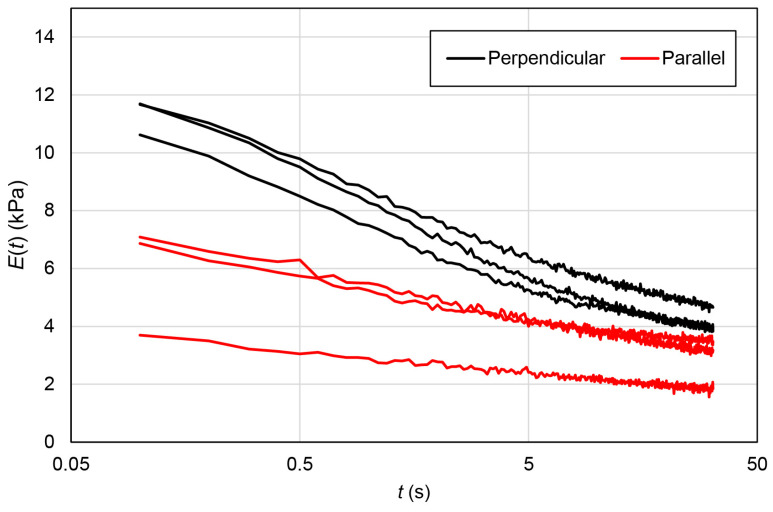
Relaxation moduli of the EMT in the perpendicular and parallel directions. All measured results are summarized in the figure (*n* = 3).

**Figure 7 gels-09-00141-f007:**
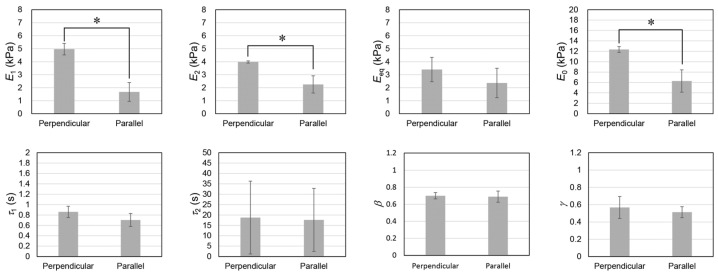
Comparison of the relaxation parameters between the perpendicular direction and the parallel direction. Error bars indicate standard deviation (*n* = 3). Asterisks indicate significant differences of *p* < 0.05. The comparisons were performed by using Student’s *t*-test.

## Data Availability

All measurement data obtained in this study are available and can be downloaded at https://drive.google.com/drive/folders/1cQKr_4mA84KArjfCXm4gk3415ZRMJ0-o?usp=sharing (accessed on 5 February 2023).

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
