# Peer review of "Construction of Engineered Muscle Tissue Consisting of Myotube Bundles in a Collagen Gel Matrix"

_gels, 2023, doi:10.3390/gels9020141_

Round 1

Reviewer 1 Report

In this work, the authors have constructed the engineered muscle tissue (EMT) by culturing embryonic cells in collagen hydrogels with unidirectionally aligned microchannels (MCCG). The obtained EMT exhibits hierarchical structure with considerable response to periodic electrostimulations and anisotropic mechanical behaviors. Overall, the work itself is systematic studied and interesting. However, following concerns should be addressed before final acceptance.

1.The elastic modulus obtained in Figure 5C is around 2000-4000 Pa, why does the initial modulus in relaxation tests (Figure 6) exceeds 10000 Pa? The authors should discuss that and provide more details in experimental operation.

2.The method to get hierarchical structure is interested. The final purpose to study or mimic the complicated structure of real muscle is to develop a artificial muscle with comparable or even better properties. However, the mechanical strength, stretchability and toughness is far away from real muscle or existed engineered artificial muscle, which significantly impedes the practical use of this material. Is it possible to further enhance its toughness?

3.The conclusion part is missed.

Author Response

I would like to thank you for carefully reviewing our manuscript and providing comments.  According to your comments and suggestions, we revised the manucript. The point-by-point response are shown in the followings:

1.The elastic modulus obtained in Figure 5C is around 2000-4000 Pa, why does the initial modulus in relaxation tests (Figure 6) exceeds 10000 Pa? The authors should discuss that and provide more details in experimental operation.

In this study, the stress - strain curve was determined by measuring the stress at 30 second after stepwise deformation. Therefore, the elastic modulus shown in Figure 5C should be comparable to the equilibrium elastic modulus (Eeq) that is a plateau modulus shown in Figure 6 but not to the initial modulus at t = 0 s. In addition, we revised and added the experimental operation of stress - strain curve.

2. The method to get hierarchical structure is interested. The final purpose to study or mimic the complicated structure of real muscle is to develop a artificial muscle with comparable or even better properties. However, the mechanical strength, stretchability and toughness is far away from real muscle or existed engineered artificial muscle, which significantly impedes the practical use of this material. Is it possible to further enhance its toughness?

Reproducing the fracture properties is important to apply EMT as actuators. In this study, we did not measured fracture properties of EMT. Therefore, we could not compare the fracture properties of EMT with the real muscle or existed engineered artificial muscle. On the other hand, at the stress - strain curve measurement, we found that EMT at day 33 was not broken by gently pinching with tweezers. Therefore, EMT could be durable against small deformations. By contrust, the durability at large deformations are still unknown. If the fracture properties of EMT are less than those of the real mucsle and the existed artificial muscles, we must develop the reinforcement method. We believe that improvements of cell density, conectivities of cells, and maturation of muscle fibers in EMT reinforces the toughness. In addition, there are various methods for controlling the mechanical properties of hydrogels. For example, the strength of collagen hydrogels could be reinforced by introducing double network structures with other polymer hydrogels. Therefore, the methods could be also useful  to enhance strength of EMT. We have already prepared the hybrid MCCG with the Matrigel and been investigating the effect of Matrigel on the myogenic differentiation and the mechanical properties of EMT. This will be reported elsewhere. 

3.The conclusion part is missed.

We added the conclusion part in the revised manuscript.

Reviewer 2 Report

1. Please proofread the manuscript as there are many grammatical errors and incomprehensible sentences. 

2. The Figures should be added after they were mentioned in the text. For example, Figure 1 should be after the paragraph in the Hierarchical structures of EMT section.

3. Pages 2-3, lines 69-71. How long was the culture vessel containing cells and scaffold rotated at 85 rpm? Include this information in the methodology.

4. The Conclusion section is missing from the manuscript.

Author Response

I would like to appreciate you for carefully reviewing our manuscript and providing comments.  According to your comments and suggestions, we revised the manucript. The point-by-point response are shown in the followings:

1. Please proofread the manuscript as there are many grammatical errors and incomprehensible sentences. 

I must apporogize for my poor english writing. The manuscript was checked and edited by native speker.   

2. The Figures should be added after they were mentioned in the text. For example, Figure 1 should be after the paragraph in the Hierarchical structures of EMT section.

We moved Figures after the paragraph where they were mentioned.

3. Pages 2-3, lines 69-71. How long was the culture vessel containing cells and scaffold rotated at 85 rpm? Include this information in the methodology.

All samples were rotated at 85 rpm during the entire culture period. We added this information in the materials and methods part.

4. The Conclusion section is missing from the manuscript.

We added conclusion part in the revised manuscript.

Reviewer 3 Report

Some suggestions/comments are inserted into the attachment.

Author Response

I would like to thank you for reviewing our manuscript, checking many gramatical errors, and helpful scientific comments. According to your comments and suggestions, we revised the manuscript. After applying your comments and suggestions to our previous manuscript, it was gramatically checked by a english editing service. Point-by-point responses are shown in the replies written in the attached pdf file.

Round 2

Reviewer 1 Report

The authors have addressed my concerns and I think it is acceptable now.

Author Response

We would like to thank you for your reviewing again.

Reviewer 3 Report

Some suggestions/comments are inserted into the attachment.

Author Response

We would like to appreciate again for your carefully reviewing the revised manuscript, checking scientific expressions and providing the comments. All comments and suggestions realy helped us to improve the quality of manuscript. In addition, I forgot to replace Figure 5A-B and Figure S1A with the revised them. The point-by-point response is summarized in the attached pdf file.
